

# Satellite Mapping of PM2.5 Episodes in the Wintertime San Joaquin Valley: A "Static" Model Using Column Water Vapor

Robert B Chatfield[1], Meytar Sorek-Hamer[1,2], Robert F Esswein[1,3], and Alexei Lyapustin[4]

[1]NASA Ames Research Center, Moffett Field, CA 94035, USA

5          [2] Universities Space Research Association, Moffett Field, CA, USA

[3]Bay Area Environmental Institute, Moffett Field, CA, USA

[4]NASA Goddard Space Flight Center, MD, USA

**Abstract:** The use of satellite Aerosol Optical Thickness (AOT) from imaging spectrometers has been
successful in quantifying and mapping high PM2.5 (particulate matter mass < 2.5 μm diameter) episodes for pollution
abatement and health studies. However, some regions have high PM2.5 but poor estimation success. The challenges
in using Aerosol Optical Thickness (AOT) from imaging spectrometers to characterize PM2.5 worldwide was
especially evident in the wintertime San Joaquin Valley (SJV). The SJV's attendant difficulties of high-albedo surfaces
and very shallow, variable vertical mixing also occur in other significantly polluted regions around the world. We
report on more accurate PM2.5 maps (where cloudiness permits) for the whole-winter period in the SJV, Nov 19,
2012–Feb 18, 2013. Intensive measurements by including NASA aircraft were made for several weeks in that winter,
the DISCOVER-AQ California mission.

We found success with a relatively simple method based on calibration and checking with surface monitors and
a characterization of vertical mixing, and incorporating specific understandings of the region's climatology. We
estimate PM2.5 to within ~7 μg m$^{-3}$ rms error and with $R$ values of ~ 0.9, based on remotely sensed MAIAC (Multi-
Angle Implementation of Atmospheric Correction) observations, and that certain further work will improve that
accuracy. Mapping is at 1 km resolution. This allows a time sequence of mapped aerosols at 1 km for cloud-free days.
We describe our technique as a "static estimation." Estimation procedures like this one, not dependent on well-mapped
source strengths or on transport error, should help full source-driven simulations by deconstructing processes. They
also provide a rapid method to create a long-term climatology.

Essential features of the technique are (a) daily calibration of the AOT to PM2.5 using available surface monitors,
and (b) characterization of mixed-layer dilution using column water vapor (CWV, otherwise "precipitable water").
We noted that on multi-day timescales both water vapor and particles share near-surface sources and both fall to very
low values with altitude; indeed, both are largely removed by precipitation. The existence of layers of H$_2$O or aerosol
not within the mixed layer adds complexity, but mixed-effects statistical regression captures essential proportionality
of PM2.5 and the ratio variable (AOT/CWV). Accuracy is much higher than previous statistical models, and can be
extended to the whole Aqua-satellite data record. The maps and time-series we show suggest a repeated pattern for
large valleys like the SJV — progressive stabilization of the mixing height after frontal passages: PM2.5 is somewhat
more determined by day-by-day changes in mixing than it is by the progressive accumulation of pollutants (revealed
as increasing AOT).





## 1. Introduction

The San Joaquin Valley (SJV) is an important agricultural area, characterized by poor air quality (Figure 1). The SJV gives an example of a region with frequent air pollution episodes, challenged by difficulties as varied particle characteristics with hard-to-quantify sources from domestic burning and spatially distinct ammonia and

nitrate precursors. The 60,840 km$^2$ area (with approx. 4 million residents) is located southeast of San Francisco, between the Coastal Mountain Range to the west and the Sierra Nevada Range to the east (Sorek-Hamer et al., 2013). Previous studies in this region reported a range of correlations between satellite-borne AOT and daily/ hourly collocated ground PM2.5 measurements in this region. Using linear tools resulted in little or no correlation (Engel-Cox et al., 2004; Ballard et al., 2008; Justice et al., 2009), while applying non-linear methods improved the

correlation to R=0.71 (Sorek-Hamer et al., 2013).

More broadly, atmospheric particulate matter (PM) pollution in the respirable range, PM2.5, is recognized as a major threat to human health for some time (Brunekreef and Holgate, 2002, Dominici et al., 2006; Franklin, 2007 Kloog et al., 2013; Schwartz, et al., 1996; Zanobetti et al., 2009). Epidemiological studies have been hampered by the availability of relatively few PM2.5 measurement stations relative to the broad dispersal of populations affected.

A variety of methods have been employed to estimate exposure, e.g., proximity-based using GIS, interpolation between sparse monitoring sites, land-use regression models, line- or area-dispersion plume models, 3-d atmospheric source-and-transport models, and models using information from imaging satellites, often including also land-use regression and proximity (Sorek Hamer et al., 2016). Sparse PM2.5 monitoring spatial networks may limit our ability to accurately assess human exposures to PM2.5, since concentrations measured at an outdoor site

may be less representative of the subjects' exposures as the distance from the monitor increases (Bell et al., 2007; Lee et al., 2011).

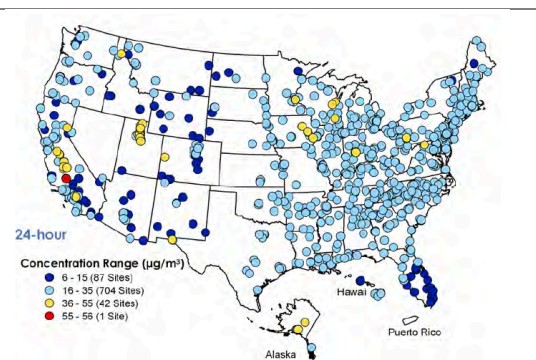

**Figure 1. Annual average PM2.5 (24-hr average) as estimated for 2010. Source: EPA. (https://www.epa.gov/outdoor-air-quality-data/interactive-map-air-quality-monitors, and RSIG, https://www.epa.gov/hesc/rsig-data-inventory )**

For this reason, there has been extensive development of techniques to make best use of satellite-borne optical extinction as seen from moderate-resolution atmospheric imagers. Aerosol Optical Thickness (AOT) is typically

reported as a vertical column integral of extinction above the ground footprint observed. Methods using AOT to



assess exposure to PM showed early successes, but certain regions remained very poorly characterized (Engel-Cox et al., 2004, Liu et al., 2009, Gupta et al., 2006; Koelemeijer et al., 2006, Hoff and Christopher, 2009). Engel-Cox (2004) found correlations of AOT with PM2.5 for valleys along the US Pacific Coast ranged from –0.2 to +0.3, i.e., very little variance explained. MISR technology aided greatly (Liu et al., 2007), but yields mostly monthly averages

over years (van Donkelaar et al., 2010), limiting event and epidemiological analysis.

AOT may be strongly affected by particles encountered well above the planetary boundary layer and different particulate composition. In addition, cloud cover severely limits the actual spatial coverage of AOT (Ford and Heald, 2016). Yet, in spite of these limitations (Jin et al, 2019), AOT has been employed extensively for assessing PM concentrations (e.g. Liu et al., 2018, Franklin et al., 2017; Van Donkelaar et al., 2015, 2016; Kloog et al., 2015,

2014; Hu et al., 2014; Sorek-Hamer et al., 2013; Hoff and Christopher 2009).

In regard to the SJV, considerable work has been published, since it was the site of two major intensive studies, CRPAQS (California Regional PM10/PM2.5 Air Quality Study, Chow et al,2006) and DISCOVER-AQ California (Deriving Information on Surface Conditions from COlumn and VERtically Resolved Observations Relevant to Air Quality, https://www.air.larc.nasa.gov/missions/discover-aq/discover-aq.html, more references below and at web

site). There was a very useful analysis of particle composition for a well-instrumented Fresno surface site for this period (Young et al., 2016). This study added detail to the Watson and Chow (2002) analysis of an earlier intensive study of the area; in particular, the striking dominance of nitrate and organic aerosols in a regular diel pattern. Watson and Chow reference several publications describing that intensive. Johnson et al, (2014) made a three-dimensional modelling study of methane emissions that also helps describe the mixed-layer of the specific

DISCOVER-AQ period). Lidar gives a very helpful view of complexities of submicron particle abundance and properties within the mixed layer and the uniformity of the mixed layer top (Sawamura et al., 2017).

Application of modelling with satellite AOT columns from different satellite platforms for the DISCOVER-AQ (included within our study period) was able to achieve $R^2$~0.8. These results were achieved for just the DISCOVER-AQ period of ~6 weeks and with separate sub-regions of the Central SJV. They highlight the

complexity of composition and source-driven simulation (Friberg et al, 2018). The Friberg publication is highly recommended as a comparison to this effort, and has extensive references regarding the SJV and the details required for source-driven modelling.

There are several related goals in producing PM2.5 maps and assessing their accuracy. The Friberg et al., work primarily aimed to constrain CMAQ downwind of the surface air quality stations, and in particular, to constrain

particle type as much as possible, along with concentration, using MISR constraints (Ralph Kahn, personal communication 2019). Our goal was to produce a large set of maps characterizing one winter in a particular setting, inland Mediterranean valleys, with the aim of allowing air pollution professionals to understand particulate episodes and to improve sources and simulation details (e.g., transport error) for source-driven models. Goals of the Dalhousie group are to improve annual average exposure: they see that as the principal driver for health effect (Van

Dankelaar et al., 2010, 2015, 2016). A main goal of NASA's MAIA (Multi-Angle Imager for Aerosols) mission is similarly deliver new data for a each-day mapping of PM2.5 exposure sufficient for full studies of health effects (Diner et al., 2018, https://maia.jpl.nasa.gov/). In pursuit of that goal for the MODIS Aqua dataset, we will indicate some preliminary, meteorology-based ideas for estimating high aerosol concentration when clouds prevent the use of remote sensing data.



Due to the complex meteorology of the San Joaquin flows and uncertainties surrounding the sources of
ammonia, nitrogen oxides, and residential-burning smoke, we attempt to separate out some certain aspects of
complex 3-d source-driven modeling (Friberg et al. 2018, and references) with a "static model" which does not
attempt to simulate transport, but rather uses observational records related to vertical mixing and AOT. The spatial
maps produced can give a more detailed check on the 3-d process modeling. They also allow the whole MODIS

record of winters to be analyzed efficiently so as to reveal patterns and trends. We emphasize this and further
extension the interpretation of satellite radiances, attempting to remain close to physical interpretations by using
both MAIAC AOT and CWV retrievals. MAIAC Column Water Vapor (CWV) (Lyapustin et al., 2018) retrievals
have been quite acceptability validated with the AERONET CWV measurements in higher CWV environments
(Martins et al., 2017, 2018). It has not been previously recognized as a tool for improving ground PM estimation and

in particular, in the SJV.

## 1.1   Data

MAIAC AOT and CWV

The Multi-Angle Implementation of Atmospheric Correction (MAIAC) is an operational algorithm developed
for MODIS Collection 6 (C6) data (Lyapustin et al., 2011a,b). This algorithm applies a dynamical time series
technique to derive the MODIS surface bidirectional reflectance factor and atmospheric retrievals at a 1 km
resolution, such as AOT, and CWV (Lyapustin et al., 2008; 2011b). MAIAC AOT retrievals present an expected
error within 15% and relatively good correlation coefficient (R) with AERONET measurements in the study area

(Lyapustin et al., 2018).

MAIAC data has been used from both Terra and Aqua satellite with a daily overpass at ~10:30 and ~13:30 UTC,
respectively. Data has been obtained for the period of Winter 2012-2013 (November 2012-April 2013).

AERONET AOT and CWV

AERONET (AErosol RObotic NETwork) is a global network of automatic sun-and-sky radiometers for aerosol
monitoring (Holben et al., 1998). Direct sun measurements are used to compute the AOT values at seven
wavelengths (340, 380, 440, 500, 675, 870, 1020 nm), while CWV retrievals are derived from the channel 940 nm
(Schmid et al., 2006). The AERONET data were obtained for the study period with cloud screened and quality-
assured at V3 Level 2 products. The AERONET AOT values were interpolated to a 550 nm using quadratic fits on a

log–log scale. Details on instruments and monitoring sites of the DISCOVER-AQ campaign are available at:
http://www.nasa.gov/mission_pages/discover-aq/instruments/index.html. Archived DISCOVER-AQ data are
available at the NASA LaRC Science Data for Atmospheric Composition website: http://www-air.larc.nasa.gov/
index.html.


Ground PM2.5 Concentrations



Hourly ground PM2.5 concentrations were obtained from the USA Environmental Protection Agency (EPA) at +-60 minutes from the satellite overpass. Data were obtained from stations that reported non-negative PM values over the whole study period. (https://aqs.epa.gov/aqsweb/airdata/download_files.html#Raw)


PBL

Momentum-based PBL depth, 10-m wind, and some CWV quantities for the model were taken from the archive of the NOAA Rapid Update Cycle (RAP) model available for this period. (Choice of MAIAC or RAP estimates is discussed later. The model archive had a nominal 13 km resolution and a temporal resolution of one

hour, so that model quantities could be matched closely to the satellite overpass times. Unreported examination of the AERONET data for the period suggested that the temporal resolution of the MAIAC AOT was quite accurate. The HSRL2 aerosol data as described by Sawamura et al., (2017) suggested that depths of afternoon mixing tops were adequately described by a 13 km resolution model, as did adjacent spirals of the P3-B as described by Michael Shook (Shook et al., 2013) .AOT could however vary on relatively short distance scales, e.g. within 0-2 km of

roadways when winds were parallel to the road. We shall see the consequential variations in estimated PM2.5 later in the processed results.

## 2.  Motivating Meteorological Perspective

Koelenmeijer et al, 2006 give a succinct description of the relationship between AOT and dry particle mass.

We adopt their simplification describe the relationship of AOT to PM2.5 using a simple equation where all particles are idealized as evenly mixed throughout a layer mixing to sensors near the ground, and the thickness of the mixed layer is $\Delta z_{ML}$

$$PM2.5 = f(\text{AOT}) = \frac{\text{AOT}}{\Delta z_{ML} \cdot M(Composition, RH)} \qquad \text{Eq. 1}$$

The factor in the denominator, M (for "magnification") describes the relationship of the optical extinction to "dry particle mass smaller than 2.5 μm aerodynamic diameter" which is the motivated definition of PM2.5. (PM2.5 also has a definition by a "Federal Reference Method" which is formulated to approximate the physical definition as closely as possible.) The factor $M$ then is a function of particle composition and the extinction coefficients $b_{Ext}$ associated with the components, one of which may be largely absorbed water. Particle composition and ambient

relative humidity, RH, then interact with each other to determine the water content. It is significant that RH is a function of temperature and therefore altitude, with highest RH at the top of a well-mixed layer.

This work emphasizes and attempts to exploit features of regional aerosol haze palls that parallel features of aerosol mass and a different measure of water vapor.

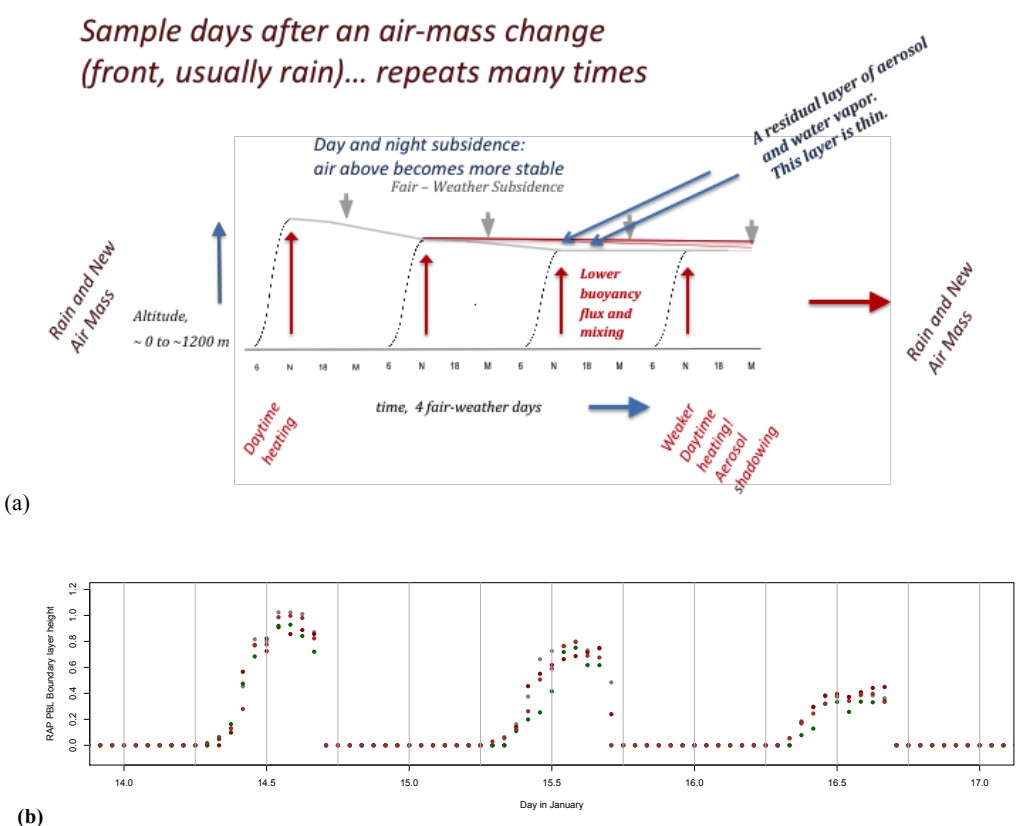

(a)

(b)

**Figure 2. (a) Conceptual figure describing the fair-weather PBL top for successive days in a clear-weather PM2.5 episode motivating this study. See text. (b) Simulation in the RAP model of planetary boundary layer height for momentum at three SJV PM2.5 stations. Periods from 11 AM to 3:30 PM approximate the mixed layer for that period and time following, although advection may change the concentrations mixing to that height. Maximum PBL-top altitude may not be accurate for the station, but shape of diel profile is appropriate.**


      Figure 2 illustrates a conceptual idea of the fair-weather simulation we focus on. Both regional particulate pollution and water vapor originate from the Earth's surface. Each tends to create relatively well mixed layers over several days, transported most significantly by a repeated daily cycle of mixing. The mixing of momentum is most active from just before noon through the mid-afternoon, creating an afternoon mixed layer, and water vapor and

aerosol most typically mixes well up to this layer. Turbulent mixing depths vary from day to day, and these can create lofted layers of pollution cut off from the surface on the day of AOT and CWV observation. Flows in the San Joaquin can be greatly influenced by the nearby mountains, with flows day and night promoting some upslope transport of material which can recirculate, detached from mixing on following days. Consideration of subsidence of air into the San Joaquin mixed layer suggests a flow-through time for aerosol and water of 2–3 days for some

situations (Caputi et al., 2018). Mixing of entrained and mixed layer air allows for continued accumulation of pollutant aerosol in the valley as Figure 2 shows.

Particles and water vapor are emitted and accumulate in the same region, and they are mixed similarly each midday and afternoon by convective stirring. The height of mixing can be determined by variations in the buoyancy flux from the surface and varying vertical subsidence velocities, responding to larger scale weather patterns, during successive days. Figure 2 does not show the effect of particle transport or water vapor transport for a specific location, but the PBL top, which is strongly controlled by local heat and water vapor fluxes at the surface.

If the mixing height is lower on succeeding days, then any water vapor and any particles at the top of the mixed layer are trapped in an "elevated layer" which does not mix to the surface. Other common ways in which elevated layers can be formed are mixing along the side of the valley (small -scale anabatic and katabatic winds) and by differential transport, i.e., wind shear. Fires, power plant plumes, and long-distance synoptic transport can form layers that are quite separated at higher altitudes in the troposphere. Eventually there is removal of both species. Wet removal of particles is particularly effective, and the specific humidity of the air is very effectively removed by the condensation accompanying cooling and rising, according to the Clausius Clapeyron equation. Similar processes then limit the vertical spread of particles and specific humidity.

### 3.   Expected Variability of the AOT-PM2.5 relationship

Water vapor molecules also accumulate in the atmosphere over a period of several days (typically a somewhat longer period), and both aerosols and water vapor are cleared from a particular place by cloud removal processes (venting, rainfall) and by airmass replacement. In the case of high pressure systems in which air pollution episodes occur, such replacement is a common feature. If the other variables are available by measurement, e.g., airplane measurement such as in DISCOVER-AQ (https://discover-aq.larc.nasa.gov/data.html), Equation 1 can be solved for $\Delta z_{ML}$, defining an equivalent mixing height for particles. Similarly, we can write equivalent mixing depth of water vapor, $\Delta z_{e\,H2O}$ :

$$\Delta z_{e\,H2O} = \left.\int_0^{Top} \rho_{H2O}(z)\,dz \middle/ \bar{\rho}_{H2O}(ML,RTP)\right. = CWV / \rho_{H2O}(ML,STP)$$

Eq. 2

where CWV is in g / cm2 , $\rho_{H2O}(z)$ and $\bar{\rho}_{H2O}(ML,RTP)$ correspond to the vertically distributed water vapor and appropriately average water density of the mixed lay. CWV is available from the MAIAC analyses yielding AOT. Making the assumption that the heights are the nearly equivalent for water vapor and aerosol, we may write

$$PM2.5 = f(AOT) = \frac{AOT}{CWV} \frac{\bar{\rho}_{H2O}(ML,RTP)}{M(Composition,RH)}.$$

Eq. 3

PM2.5 is calculated at EPA reference temperature (25 C) and pressure (1 atm), water vapor quantities in g cm–3 and $\Delta z_{e\,H2O}$ is in cm.

Work reported by Shook et al., 2018, described the vertical distribution of trace species with a vertical coordinate normalized to his estimated afternoon mixed layer top, This suggested to us that water vapor had vertical distributions that were usefully similar.  The decline of water vapor was not as sharp, often showing a rapid decrease; the drop in scattering was dramatically rapid,



This basic understanding does not fully explain the success of the mixed effects model that we observed for the San Joaquin Valley. Furthermore, analyses of the Baltimore-Washington area not described here suggest that it works more broadly. Both aerosol and especially water vapor often exhibit layers not in continual contact with surface monitors. These we will call "elevated layers." In-situ measurements on aircraft and also lidar measurements from ground lidars looking downward from aircraft (Sawamura et al., 2017), and satellite lidar (CALIPSO) reveal aerosol layers with significant optical thickness above the mixing layer. Similarly, airborne measurements in the

DISCOVER-AQ intensive measurements of 2013 suggest a fraction of water vapor lies above the mixed layer for water. Allow these portions of total AOT and CWV layers to be quantified as AOTe and CWVe ('e' stands for elevated). There can be several individual layers. AOTe and CWVe refer to the total amounts of extinction and water vapor mass. Thus there is an approximate equation upon which to base regression estimation:

$$\text{PM2.5} = \frac{(\text{AOT} - \text{AOTe})}{(\text{CWV} - \text{CWVe})} \frac{\bar{\rho}_{H2O}(ML, RTP)}{M(Composition, RH)}$$

Eq. 4

We found in ensuing work that approximating $\bar{\rho}_{H2O}(ML, RTP)$ by $\rho_{H2O}(z = 0, current\ conditions)$ added only a small amount to the variance explained by the regression given other limitations of the approximation. (Possibly relative humidity effects or the correlation of water density with temperature could be complicating correlated factors.)

     We calibrate the relationship $f(\text{AOT})$ using data at official PM2.5 stations, and make the calibration daily. It is

our observation that $f$ varies only over a small range when there are several MODIS observations on the same day, and that it varies in a limited way between neighboring stations in a local region. The definition of "region" is based on that similarity, and it suggests similarity of $\Delta z_{ML}$ and M, i.e., similar aerosol characteristics and boundary layer behavior. This similarity does not apply when the wind shifts greatly between times or between stations, e.g. when a front passes. Fortunately for our understanding of pollution episodes, frontal passage days tend not to have high

PM2.5 .

     We distill these understandings when we formulate a regression equation

$$\text{PM2.5}_{i\,s} = (a + \beta_i)\,(\text{AOT}_{is}/\text{CWV}_{is}) + \alpha_i + \varepsilon_{is}$$

Eq. 5

where the subscripts i describe "instance" or calendar date, and the subscripts s describe "station," so that AOT and PM2.5 form a two-dimensional table. While elevated layers of water and aerosol are common, we will see that it appears that this regression equation allows rather good fits. This can happen when AOTe << AOT and CWVe <<

CWV to a sufficient degree, or else when there are approximate linear (slope + intercept) relationships obtaining between the numerator and denominator of Equation 4. Essentially the terms are absorbed into constant parameters for the day, $\alpha_i$ and $\beta_i$, along with other parameters like M. AOTe and CWVe are considered to be essentially constant over the region. In fact, this degree of constancy can be taken to define the "region" of application.

     We may these terms into constants $\alpha_i$ and $\beta_i$ works under an implicit assumption of uniformity in AOTe and

CWVe throughout the region, or at least a uniform linear dependence with AOT and CWV.

     Given the independent nature of i and s, the regression must be solved by "mixed effects" methods described below. The subscript s need only be independent of i, so later we will use it to denote "situation" or the hour of the day when there are many observations made at one station on one day i. It is not assumed that the consecutive order





of the day observations necessarily describe any continuity in f. Observations show that there is often continuity, but
that the continuity is quickly broken when frontal passages or rain affect the region.

Writing Equation 3 in the form used for mixed-effects models, we separate a general term from the terms that depend on i or calendar date.

$$\text{PM2.5}_{is} = a \cdot \text{AOT}_{is}/\text{CWV}_{is} + c + (\alpha_i + \beta_i \cdot \text{AOT}_{is}/\text{CWV}_{is}) + \varepsilon_{is} \qquad \text{Eq. 6}$$

A commonly used shorthand is the Wilkinson and Rogers (1973) form, accepted by many software packages,

$$\text{PM2.5} \sim \text{AOT}/\text{CWV} + (\text{AOT}/\text{CWV} + 1 \mid \text{DOY}) \qquad \text{Eq. 7}$$

where DOY describes the calendar date subscript i. This formalism also describes the columns of the regression
matrix to be solved.

It is tempting to generalize this relationship to recognize that there is often correlated behavior between stations, but with some constant offset

$$\text{PM2.5}_{is} = a \cdot \text{AOT}_{is}/\text{CWV}_{is} + c + (\alpha_i + \beta_i \cdot \text{AOT}_{is}/\text{CWV}_{is}) + \gamma_s + \varepsilon_{is} \qquad \text{Eq. 8}$$

However, if one allows such variations at monitoring stations, it can be difficult to decide what values of $\gamma_s$ to use between stations. This is an attempt to describe "sub-regionality," that is similar behavior within a region
modified by slight and geographically coherent variations which allow spatial interpolation.

For those not familiar with mixed-effects models, we mention that the procedure is similar to the use of dummy variables, where coefficients $u_i$ multiply a set of discriminating variables, equal to 1 when i takes on the value of a particular instance/day, and 0 for all other instances. The mixed-effects techniques similarly solves a much larger regression equation, but has better theoretical development. Note that the number of observations is Ni
times Ns, while the number of parameters is linear in Ni and Ns, where the N's signify the number of each. When Ni and Ns > 5, the problem becomes increasingly over-determined.

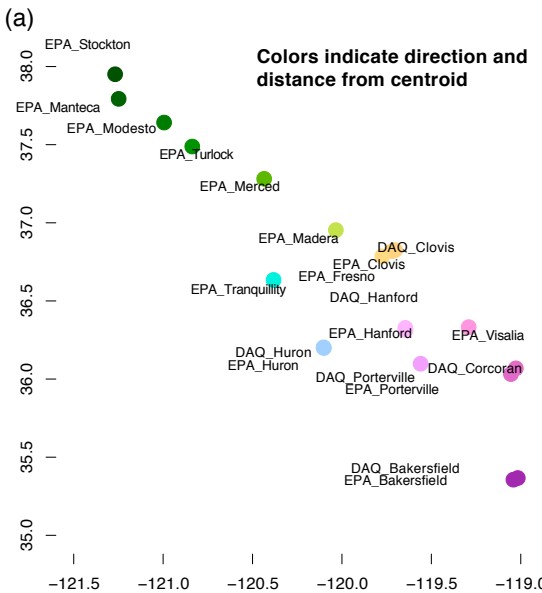

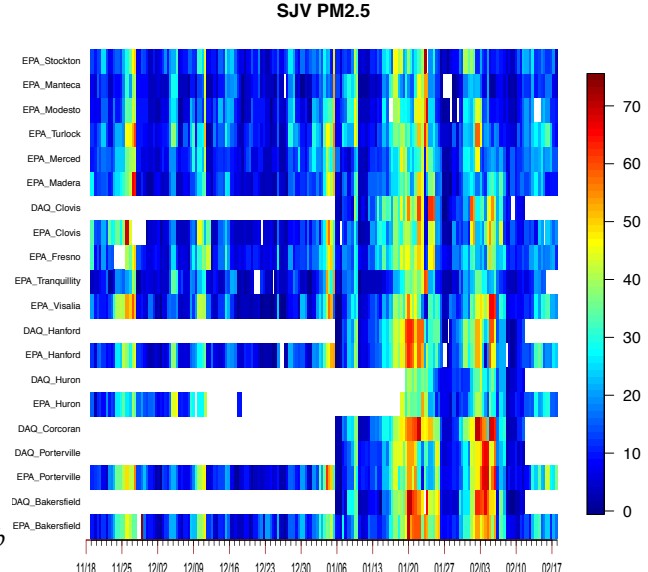

**Figure 3. (a) Locations of stations in the SJV used; color coding allows nearby stations to be identified. (b) Matrix plot of PM2.5 at the sites for the period November 19, 2012 through February 18, 2013. (b) Another view that summarizes the variability of observed PM2.5 is shown in Figure 4(a).**

## 4.      Observations and an Overview of Pollution Episode Trends

In this section, we will show how components of a mixed effects model that utilize CWV contribute to its explanatory power. We examine the relationships and predictive ability for PM2.5 observed at SJV measurement stations for the winter season encompassing high-pollution periods, Nov 19, 2012 to February 17, 2013. Stations from Bakersfield in the South to Stockton in the North were included. The figure shows several episodes affecting most of the Valley; one period with more stations reporting includes the DISCOVER-AQ period. This period has

additional aircraft data which motivated this work, but are too lengthy to describe in this publication.

Figure 3 (a) shows the locations of all stations used in this work; the stations include much of the Valley from Stockton to Bakersfield. Some of the stations labelled DAQ were in operation only during the DISCOVER-AQ California period. A color wheel was used to assign colors to the graph; this allows identification of stations' latitude, longitude, and proximity in later graphs comparing observations and our fitted values. Figure 3(b) describes

the rise and fall of PM2.5 pollution using the station reports. The rows represent stations and are arranged north to south. Several major episodes are immediately seen, as well as differences in their intensity and timing of development. The DISCOVER-AQ observations were limited to the period shown, January 8th through February 10th, 2-13. Differences between the PM2.5 values observed at nearby stations, one DISCOVER-AQ, one California





Air Resources Board (labelled "EPA" for the dataset origin) give an impression of local variability; differences

between observations at Clovis are quite apparent.

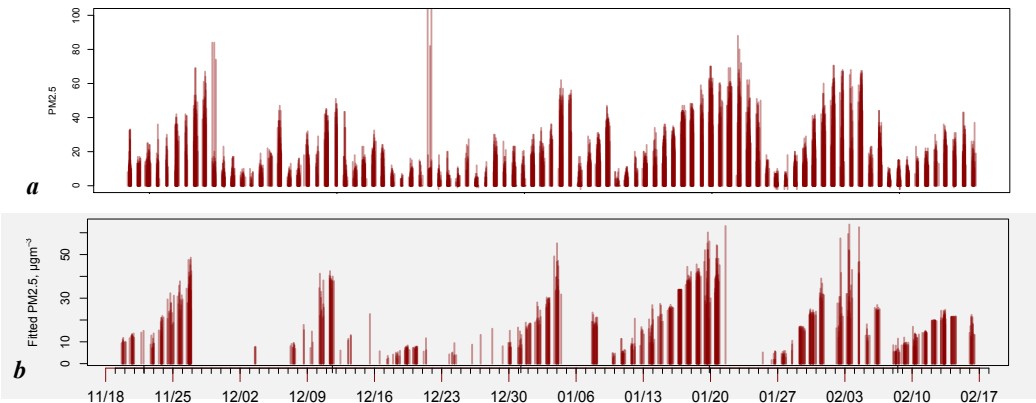

**Figure 4. (a) PM2.5 as observed at all stations for the winter period extending from November 2012 to March 2013. The graph has vertical bars drawn with partial transparency, so that careful inspection of a single day describes all the observations in the Valley for the day. (b) PM2.5 as fitted by the regression with slopes and intercepts, described further below.**

**Table 1. Comparison of results using different terms in the mixed effects model**

| Wilkinson Shorthand | Terms | $R$ | Remaining RMS Error |
|---|---|---|---|
| ~ AOT | $\text{PM2.5}_{i\,s} = a \cdot \text{AOT}_{is}$ | *0.40* | 14. |
| ~ AOT/CWV | $\text{PM2.5}_{i\,s} = a \cdot \text{AOT}_{is}/\text{CWV}_{is}$ | *0.48* | 13. |
| ~ (1 \| DOY) | $\text{PM2.5}_{i\,s} = c + \alpha_i + \gamma_s + \varepsilon_{is}$ | *0.78* | 10. |
| ~ AOT/CWV + (AOT/CWV – 1 \| DOY) | $\text{PM2.5}_{i\,s} = a \cdot \text{AOT}_{is}/\text{CWV}_{is} + c + \beta_i \cdot \text{AOT}_{is}/\text{CWV}_{is} + \varepsilon_{is}$ | *0.88* | 7.44 |


| ~ AOT/CWV + (1 \| DOY) | $PM2.5_{i\,s}$ $= a \cdot AOT_{is}/CWV_{is} + c$ $+ \alpha_i + \gamma_s + \varepsilon_{is}$ | *0.85* | 8.03 |
|---|---|---|---|
| ~ AOT/CWV + (AOT/CWV +1 \| DOY) | $PM2.5_{i\,s}$ $= a \cdot AOT_{is}/CWV_{is} + c$ $+ (\alpha_i + \beta_i \cdot AOT_{is}/CWV_{is})$ | *0.90* | 6.72 |
| ~ AOT/CWV + (AOT/CWV +1 \| DOY) + longitude | $PM2.5_{i\,s}$ $= a \cdot AOT_{is}/CWV_{is} + c$ $+ (\alpha_i + \beta_i \cdot AOT_{is}/CWV_{is})$ $+ \gamma_s + \varepsilon_{is}$ | *0.91* | 6.44 |

## 5.        Which Information Contributes to PM2.5 Maps

Using the MAIAC 1x1 km estimates for each day for the location of each aerosol monitoring station and the PM2.5 measured at overpass time for that day, we may solve the estimation equation, Eq. 6. The complete simulation of PM2.5 measurement at all stations where MAIAC data allowed is shown in Figure 4(b). The technique can be used for all years and the whole area of the SJV where MODIS data is available. We used the complete model as described  in Equation 6, "slopes and "intercepts", but without

any time-independent spatial variation allowed ($\gamma_s$ ).  Three features deserve immediate comment. First, there are patterns of gradual increase of PM2.5 up to 45–80 µg m$^{-3}$ followed by relatively sudden decrease to levels near 5 µg m$^{-3}$. Second, the regression technique using AOT / CWV, as estimated individually for each day, captures the variation rather well, for all days where estimates can be made. Individual, exotic high values are not captured. Third, there is a pattern where the end of an air pollution

episode, showing very high values, is not captured by the technique. These are simply days where MODIS observations were not available, almost always due to cloud cover. We expect that these are readily explained in terms of weather phenomena especially typical of the American West during wintertime. Pollution episodes are ended with the approach of warm fronts with high clouds, followed in a few days by the cleansing effects of rain, air mass replacement, and higher wind. We will return to this

topic later.

        To understand what information is used by the technique and how important is that information, based on the series of regression estimates we present; we argue that there is a cumulative aspect to explanation. For example, when we include one statistical variable, e.g. $\alpha_i$, describing variation by day but constant for all stations of the day, then the regression with AOT/CWV becomes much more informative. A general relation describing the slope of

PM2.5 with AOT/CWV becomes more useful when an appropriate intercept (offset) is provided.



Consider both Table 1 and Figure 5 as they describe cumulative effects of adding information. First, we note that AOT alone is not very informative about PM2.5. This would seem to follow naturally from Equation 1, since variations in mixing depth and composition are not considered. Figure 5a show many station observations with high PM2.5 but low AOT, and vice versa. Slight but significant improvement is made when column water vapor, CWV, is introduced to provide some information on mixing depth and dilution. *R* improves to 0.48 but the remaining error is nearly as high. Still, some linear relationship begins to show for perhaps 60% of the data.

A side comment regarding significance: *R* and remaining RMS error (in $\mu g\ m^{-3}$) are shown in Table 1. We also performed two other tests not tabulated. An analysis of the Kuhlbach-Liebler divergence (Hastie et al., 2009), where possible, suggested each successive test in the table clearly adds information regarding PM2.5. The number of observations justified the number of additional parameters. While the numerical values are difficult to compare to other examples of regression, they show similar trends as *R* and RMS error, i.e. accuracy becomes increasingly hard to improve as *R* increases. Another test was leave-out-one cross validation (Hastie et al., 2009). Each individual station was omitted, and the regression based on the remaining stations was tested against observations at that station. The cross-validated mean squared error was about 7.8 $\mu g\ m^{-3}$ at most for the most informative regressions shown.

Now consider a popular alternative to the use of satellite data. The third regression shown estimates of satellite data to particulate estimation, this has been shown to surpass, or at least approximate the only $\alpha_i$, i.e, assign a single PM2.5 estimate for each station based only on the individual day. Color-coded maps of PM2.5 drawn for a region have a single color which varies from day to day. In many applications of satellite data to particulate estimation, this has been shown to surpass, or at least approximate the results of use of AOT (Sorek-Hamer et al., 2017). $R \sim 0.78$, RMS error $\sim 10\ \mu g\ m^{-3}$. Its success emphasizes the regional similarity of conditions defining PM2.5 concentrations, and their extensive spatial correlation. An explanation is that respirable PM2.5 is defined by daily weather and orientation to major sources.ß

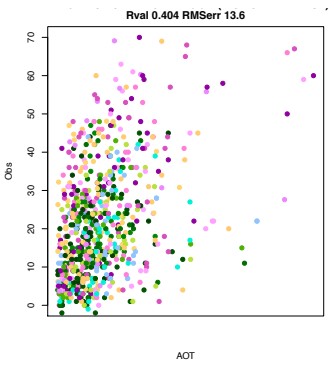

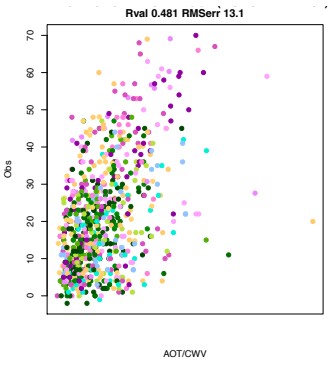

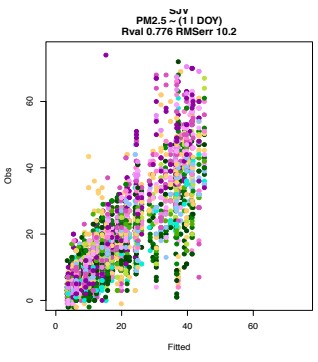

*a*                         *b*                         *c*





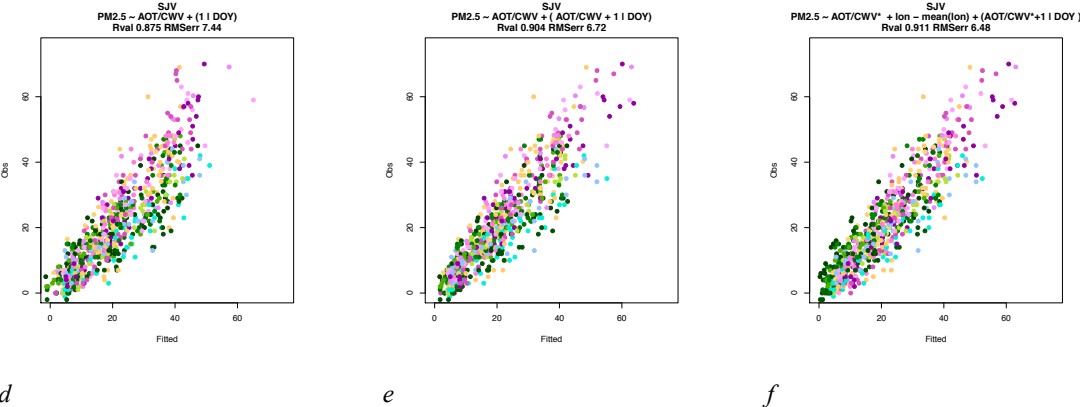

d                                       e                                       f

**Figure 5. Progressive improvement of PM2.5 simulation showing the roles of daily calibration and AOT/CWV descriptions of aerosol vertical dispersion. Station observations (µg m$^{-3}$) are shown on the y axis, estimators on the horizontal. Note the progressive refinement of *R* and remaining rms error. See text. (a) Use of AOT only, an early methodology. (b) Some improvement using AOT/CWV but no daily calibration. (c) More improvement with daily calibration (mixed effects using intercepts $\alpha_i$) (d) Clearly improved linearity when combing intercepts with AOT/CWV (e) Estimating daily "random" intercepts and slopes improves RMS error and R. (f) A simple description of variation within the region (longitude) aids the estimation slightly (RMS error ~ 6.48 µg m$^{-3}$, *R* ~ 0.91)**

Once the regional similarity of pollutant conditions is recognized, it becomes appealing to combine information. The fifth estimate shown below does just this and shows a notable increase in R, 0.88, and decrease in RMS error, 8.03. This is an approximately 50% decrease in error variance. In our situation, satellite data looks to be useful. The scatterplot of Figure 4c suggests distinctly more linear behavior.

An appealing alternative is to estimate only slope variations, $\beta_i$. This is nearly as useful as estimating just $\alpha_i$, *R*

~ 0.85 RMS error ~ 10 µg m–3. Each is useful. Do the two parameter estimations give distinct information?

Estimation of varying offsets $\beta_i$ and sensitivities $\alpha_i$ does indeed help, reducing the variance by another 10%. Combining the use of AOT, CWV, and individual daily intercepts and slopes yields *R* ~ 0.90 and RMS error ~ 6.72 µg m$^{-3}$. Nevertheless, Figure 4e shows that certain stations have persistent deviations from the general swarm of points, Tranquility (pale green) is predicted high and Porterville and neighbors (red), are predicted low.

This analysis of residuals suggests that there may be spatial variations that can be specified for our stations, $\gamma_s$, but are general enough that they can be extended to maps. For this publication, we attempted a very simple variation, an east-west variation (longitude). This did improve the scatterplot for most stations, especially when considering values above ~10 µg m$^{-3}$. RMS error decreased slightly to 6.48, and the R estimate also rose slightly, to 0.911. These changes are close to the range of sample variability. The maps shown in Figure 6 also show more

convincing (subjective) agreement in magnitude and pattern. Nevertheless, many of the highest observations are underestimated by about 20%.



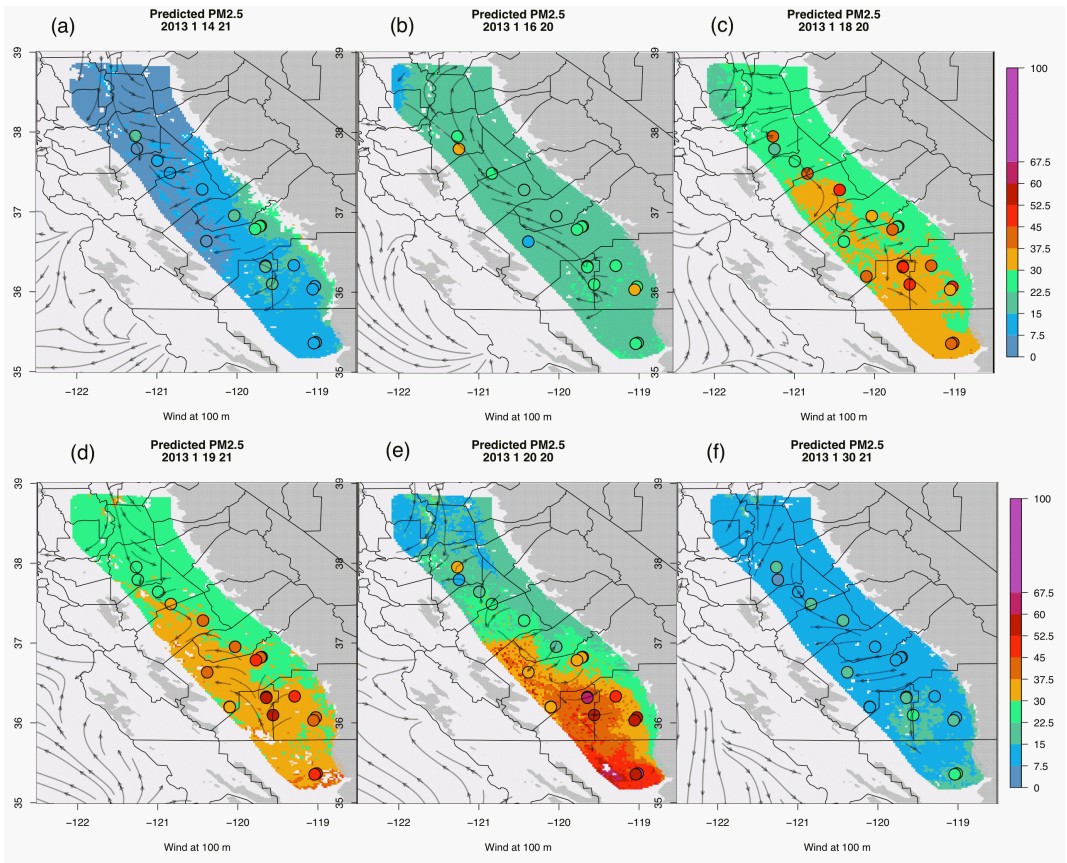

**Figure 6. Estimated surface PM2.5 at 1 km indicated overpass times for the first wintertime episode in the San Joaquin Valley. Winds at 360 m agl are also shown. Estimated RMS error is 7 µg m⁻³ with a similar limit of detection. Filled circles show station PM2.5 . In this episode, the E-W correction based on the full dataset appears inappropriate, lowering mapped estimates in the east Valley. Error should decrease with improved understanding of geographic variability.**

We used CWV rather than the RAP planetary boundary layer height for momentum, 11 to 15:30 local time. This was available in the 2012-2013 winter at times within a half hour of overpass time, However, this PBL height is not always recorded in the high-resolution RAP archive. We compared a regression very similar to the most detailed regression of Table 1, but using this PBL height. The formula used was

$$\text{PM2.5}_{i\,s} = a \cdot \text{AOT}_{is}/\text{PBL}_{is} + c + (\alpha_i + \beta_i \cdot \text{AOT}_{is}/\text{CWV}/\text{PBL}_{is}) + \gamma_s + \varepsilon_{is} \qquad \text{Eq. 8}$$

With this, the $R$ value was 0.917 and the RMS error was 6.25 µg m⁻³; these are only very slightly better than $R$ of 0.912 and the RMS error was 6.43 µg m⁻³



## 6.     Results: Maps of Estimated PM2.5

The major purpose of this work, viz. to combine AOT, CWV, and daily calibration in order to allow maps of estimated PM2.5 for all regions where MODIS can provide optical thickness data. Results using the full model with

$\alpha_i$, $\beta_i$, and $\gamma_s$ are shown (Figure 5f). Out of the 42 days in the calibration set, we consider 6 days of single major air pollution episode during middle of January, 2013, a period that was largely sampled by the DISCOVER-AQ ground and airplane samples. Detailed comparisons of the DISCOVER-AQ data would expand this work beyond a manageable size; such analysis is desirable. Winds are shown with streamlines and are obtained by interpolation from the RAP wind analyses.

We created 39 maps, six of which are shown in Figure 6. Accuracy is good. Residual Mean Squared Error, RMSE, ~7 µg m$^{-3}$.  This dictated the 5 µg m$^{-3}$ contour colors used: similar colors or neighboring colors show expectable agreement. Winds at 360 m for the hour of sampling have been superimposed on the maps.

There follows the description of just of one episode: On January 14, 2103, the Valley is clean (see also Figures 3 and 4). By January 16, 2013, light regional haze is accumulating, and the winds and mapped levels suggest some

accumulation towards the south. On January 18, 2013, winds have veered: in the central Valley, pollution accumulates towards the east; in the south, transport is towards Bakersfield. On January 20, 2013, winds press the accumulating PM2.5 back towards the more populated east Valley. Several days following have increasing clouds (no maps). The first day, with advancing clouds overhead but no low clouds, no front, nor rain, retains high PM2.5 at the monitors. This pattern is seen for several wintertime pollution episodes in this region. When the clouds clear,

the Valley is as clean as 14 Jan.  In the maps for January 18, 19, and 20, the maps underestimate the highest values of PM2.5 by about 20%, as noted above.



**Figure 7. Time evolution of PM2.5 and related variables for ~8 intensifying particulate episodes during the winter of 2012–2013. Dots and vertical bars indicate variable values at individual stations when available. Blank regions reflect periods of cloud cover. (a) PM2.5, μg m–3, as observed at stations (all dates) and (b) fitted PM2.5 on days and locations when MAIAC was available. (c) MAIAC AOT, 11:30 to 15:30 sun times. Note that increase is less pronounced than PM2.5, and varies between episodes. (d) CWV in g cm–2 or "cm of liquid water." (e) PBL height for the noon-**



**afternoon observations in this dataset. Morning PBL heights are much lower. (f) Ratio of CWV to PBL height (cm(H2O l) / km) , showing relative constancy over several days  CWV time series resemble PBL height graphs.**

### 7.        Intensification of PM2.5 Episodes: Pollutant Accumulation vs Confinement

The well performing mixed effects models (equations 5 and 6) led us to examine the repeated development of air pollution episodes to a maximum, striking patterns seen in Figures 3, 4, and 6. How did the independent values for various models in Table 1 vary within episodes and between episodes? Our description of the development leads to some answers in Section 7.

        Figure 7a and 7b describe the development of the episodes.  The time series of observed PM2.5 and fitted
PM2.5 are repeated from Figure 4. The times with no data are essentially cloudy times. After periods of cloudiness, particulate values typically rise until the next period of clouds. There are 7–8 such periods of rising, or weather episodes.  ("Episodes" can also refer to periods of highest particulate matter.) High values typically remain for 1–4 days after cloud obscuration. Figures 7c and 7b show the values fitted by our mixed effects regression and the values that are available for fitting. The time sequence as well as the magnitudes are in expectable agreement., but the
variability between stations is smaller on some occasions (e.g., 1/17 and 02/15).  Figures 7d shows that the time series of the ratio AOT/CWV develops from day to day as PM2.5 does, but suggests that these are modulated by differences in amplitude between weather episodes and sometime over  several days within the weather episodes, e.g., 01/13 to 01/17 and  02/04 to 02/08.  These explain the low overall correlation. In contrast, AOT shows little resemblance in the time series. Cloud water vapor, CWV, shows some tendency to decline during weather episodes
(Figure 7e) notably the values at differing stations are more similar than those for AOT. Regionwide similarity in CWV within and above the afternoon mixed layer is an appealing explanation. Note the limited variability of the ratio CWV/PBL over 3–5 days and between stations. The afternoon PBL height itself is shown in Figure 7g.  Note that it is often very low at the end of a cloudy period and then rises to high values ~1 km at the end of the cloudy period. We suggest that this reflects overcast skies and very limited convective mixing followed by rain and the
introduction of new air masses with deeper mixing of water vapor in a less stable atmosphere.

        Figure 7 describes differing causes of repeated PM2.5 buildup during cloud-free weather episodes. Progressive restriction of vertical mixing during clear-weather episodes acts to concentrate the effects of accumulated and recent pollution sources. The less stable air following a frontal passage feels increasing effects of strong subsidence, diminishing the mixing height. The threefold reduction in PBL height during major episodes, Figure 7d, nearly
matches the 4-fold increases in PM2.5 during these periods (Figure 7a). MAIAC AOT shows variability between stations, and is reflected in local PM2.5. Winds redistribute particles and AOT. Figure 7 does not make clear the fate of aerosol, but it likely escapes with mountainside winds along the Valley. The entire set of maps suggests a flow to the south and stronger outflow near the Tejon pass east of Bakersfield. These mountainside winds likely may facilitate water vapor and aerosols escape the prevalent mixed layer.

This suggests a typical behavior for the San Joaquin and similar regions in winter. A cloudy disturbance (new air mass, rain, wind) stirs the lower troposphere. This initiates a high PBL mixing on the first clear days. Typical fair-weather subsidence begins. The surface buoyancy flux is too weak to maintain these relatively high mixed layer





tops; Afternoon PBL depths and mixed layer depths decrease day by day until a depth of 300-400 m is reached. (Figure 7d). Escape from the Valley may slow, allowing accumulation of pollution from within the region or from

upwind. This further increases the surface PM2.5. Relatively local sources add to both AOT and PM2.5, and can transport them 50–100 km downwind, occasionally from east-valley sources to west-valley pollution hotspots (the map of Figure 6d). Both subsidence and surface buoyancy flux are broad-scale weather phenomena (~300 km) , and so AOT-to-PM2.5 relationships are similar on a given day with a given history of weather. Finally, warm-frontal rain approaches the region.

An examination of HSRL2 data for the DISCOVER-AQ period (Sawamura et al., 2017) suggests that there can be considerable vertical variability of aerosol extinction; the fact that AOT tends to average the whole afternoon mixed layer allows our generalized description to hold nevertheless.

Finally, we venture some ideas for filling in afternoon PM2.5 on days when MAIAC did not allow mapping due to cloud cover. Young et al. (2016) provided a thorough microphysical and chemical analysis for just the

DISCOVER-AQ period (January 14 to February 11, 2013) and just the fully instrumented UC Davis site deployed at Fresno. Their Figure 2, panels a, b, and e suggest a meteorological plausible method to interpolate PM2.5 maps into cloud-covered days.  These should be compared to the panels a, b, and c in our Figure 7, describing observed and statistically estimated particulate mass at all stations including Fresno. PM2.5 drops to values below  10–15 µg m$^{-3}$ whenever wind speeds rise to above ~2 m s$^{-1}$ and the wind direction is from a quadrant (90 degree sector) centered

on the north-northwest direction.  Their Figure 2a also describes rainfall at the Fresno site. Particulate matter does drop by ~50% from the highest observed/estimated values at the end of the clear-sky period, and further when the winds rise to 2 m s$^{-1}$ or higher. This behavior is most clearly observed in their graphs for the period January 23– January 27. Similar behavior is observed in the period February 6–February 11, although the episode has more complex increase than in the earlier, most intense episodes. The short spike up to 80 µg m$^{-3}$ on the night of February

10 is not explained, and not reflected in the afternoon-only data of our Figure 7. Nevertheless, the averages shown by Young in Figure 2e do repeat the general observation that daily average PM2.5 and afternoon PM2.5 do tend to correlate well.  For best-estimate maps of PM2.5, we suggest that the end-of-retrieval values of PM2.5 reduce gradually over a day or two. Maps of precipitation (e.g. from radar or other analyses) allow more detail. Estimates for a region should then fall to ~7 µg m$^{-3}$ whenever sustained winds rise to > 2 m s$^{-1}$ from the NNW or > 3 m s$^{-1}$

from any direction. Such wind speeds are held to mark air mass replacement (e.g, frontal passage). These ideas remain suggestions since our analysis for a single winter may not provide enough instances. The whole Aqua MAIAC period is available, but currently beyond NASA's resources.

## 8.       Variation of Random-effects Model Parameters

The preceding section gives some background so that we may understand the parameters for the random effect model. We will discuss the full Equation 4; results with mild spatial dependence (Equation 5) are very similar. The intercept $\alpha_i$ and the slope for $\beta_i \cdot \mathrm{AOT}_{is}/\mathrm{CWV}_{is}$ are the same for each day and determine the fitted PM2.5 for the regression Equation 4. We exploit this to produce a "stork plot" like Figure 8. High $\alpha_i$ is shown by tall blue lines; high $\beta_i$ is shown as a high slope. Variation in AOT/CWV contributes ~30–70% to the estimate on almost all days.

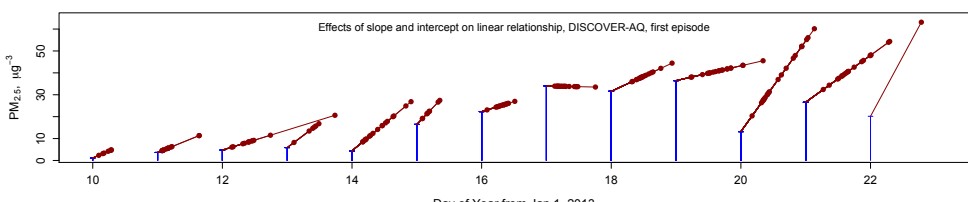


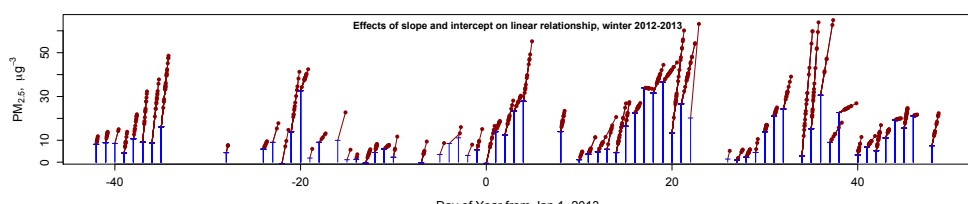

*b*

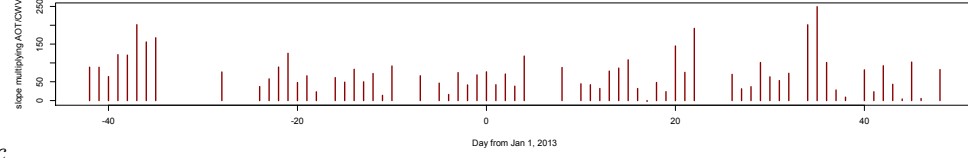

*c*

**Figure 8. Roles of slopes and intercepts in a regression fit. (a) A "stork plot" for the clear-sky air-pollution episode mapped in Figure 6. Vertical blue lines indicate the contribution of the random intercept $\alpha_i$ to the total PM2.5 fitted in**

**the model. These are the same for all geographical locations including the observation stations. The slope parameter $\beta_i$ is the same for all geographcal locations. Values of PM2.5 evaluated at the stations are shown by red dots along a line. Large values of AOT/CWV have wide vertical extent, and the corresponding high values of PM2.5 are shown as red dots at the upper right of each days plot. Highly sloped lines indicate high $\beta_i$. (b) A stork plot for the whole wintertime interval evaluates, showing several clear-day episodes. (c) the values of $\beta_i$ b vary considerably. These are shown as a time series.**

The stork plot of Figure 8a shows a puzzling progression of parameter estimates day by day. For the first days, Jan 10–14, the slope parameter accounts for the largest contribution to PM2.5. For the second part of the period, Jan 15–19, the intercept term becomes progressively more important compared to the AOT/CWV dependence. The regression equation fit (Figure 7c) has difficulty in matching the observed PM2.5 (Figure 7b) variability between stations on these days although AOT (Figure 7e) shows moderate variability around low values,

0.03–0.05. (A side note: MAIAC AOT estimates should be particularly challenged at these low values.) Then, from the Jan 20–22, the intercept contribution diminishes and the AOT/CWV dependence becomes rather larger than typical. Referring back to Figure 7e, f, and g,, these variations seem explainable: the mixed layer decreases rapidly during the first period, then reaches a minimum at ~300 m. In the last three days, the AOT increases rapidly, though the mixing depth changes little. The following weather episode is notable for high and quite variable AOT (Figure

7e), and the fitting procedure does well.


### 9. Value of Improved CWV Data

At this point, concerns about the quality of the CWV estimate should be addressed. In our analysis of the difficult San Joaquin Valley, MAIAC CWV can be frequently low compared to AERONET CWV, some error can arise from the presence of clouds in neighboring footprints. In the figures and results shown CWV was based on the MAIAC data interpolated and extrapolated where cloud-contamination made the retrieval of lower accuracy (Lyapustin et al., 2018). Figure 6 shows some small-scale variability .RAP analyses of CWV could also be used at their 13-km model-imposed width with similar results, since CWV does not vary as rapidly spatially as AOT. A better direct use of the MAIAC CWV could uses spatial averaging with a width of 3 to 6 km. Random errors in the MAIAC CWV due to the low radiances used would be reduced; considerations of source patterns suggests that CWV might not truly vary at such small scales. Improved PM2.5 values could result. We are implementing this averaging.

As understanding of MAIAC CWV improves, its role in determining daily AOT-to-PM2.5 relationships should improve; calibration of MAIAC using sun-photometer measurements can be useful in the meantime (Just et al., 2019). Note also that assimilated CWV from the National Weather Service models is constrained empirically, and so not as reliant transport descriptions as is aerosol. Here are some constraints surface-station humidity measurements constraing CWV below 0.4–1 km, thermal-radiation sounders on the GOES satellites describe water vapor partial above that; radiosonde and GPS humidity sensors give further constraint. This allows GOES AOT estimates to be used with assimilated CWV, even though GOES lacks a reflective water vapor channel (S. Kondragupta, personal communication, 2018).

### 10. Conclusions

**Goals:** We sought broadly applicables methods to estimate PM2.5 maps from satellite AOT for very polluted regions poorly described by satellite data. Ths study focusrd on the whole polluted winter season of the San Joaquin Valley (SJQ), November 19, 2012 to February 18, 2013. We sought to fulfill the overarching goal of the whole DISCOVER-AQ mission— to find general relationships between extended satellite data observations and surface air pollutant concentrations and to evaluate their success. We found success with a simple methodology that follows the meteorology of regions like the SJQ. This success recommends an approach to the remote-sensing to PM2.5 analysis, investinating important pollution regions in terms of their meteorology and sources, but carrying over methods from similar regions. For example, the Po Valley of Itally and the Northern Gangetic Plane of India may respond similary to analyses based on detailed mixing height data and related distribution indicators.

**Direct results:** We found that a combination of information utilizing (1) optical depth, (2) measures of vertical dispersion, e.g. CWV, and (3) daily calibration of PM2.5 to predictors produced significantly better quantification of PM2.5 than a competitive no-satellite-use method which we named "regional correlation" since it produces un-featured maps of PM2.5 which vary only from day to day. Our maps of estimated PM2.5 extend for all cloud-free periods November 19 2012 to February 18, 2013, essentially the whole pollution season for this winter. For that whole period, this first published attempt found good predictive value of $R \sim 0.9$ and rms error of 6.5 µg m-3. Cross-validation suggested rms error of $\lesssim 7$ µg m$^{-3}$. Analysis of residuals suggested that better rms errors could be achieved if further workallowed for sub-regioality (use of smaller regions or a geographic characterization



incorporating some spatial variation). Local variations in PM2.5 on the order of 1–3 km were noted using our method, but only when particulate accumulation could occur along-wind. Still, in order to estimate PM2.5 at $\lesssim$ 1-km scales, we expect that it will be necessary to use refined geographic information system methods (Kloog et al. 2014).

**DISCOVER-AQ comparisons advisable**: Our analyzed winter 2012z–2103 period did include the more
limited DISCOVER-AQ / California-2013 airborne-intensive study period, primarily focused on the area around Fresno. Analysis of that intensive suggested ideas (Shook et al., 2013) that motivated this work. The shorter DISCOVER-AQ period does deserve more detailed comparison to our results. Aircraft in-situ profiles of gas and particle composition, lidar profiles, very detailed surface measurements of particulate composition, and source-and-transport modeling all deserve comparison. The distribution of atmospheric particles and precursor gases is more
complex than this work might suggest. Somehow averaging appears to allow our general methods. The development of concepts and the length of this workdo not allow for such comparison. We hope that research will be encouraged.

**Usefulness of Column Water Vapor:** A major finding was that the usefulness of CWV does not become apparent unless there is daily calibration of the AOT/CWV relationship to PM2.5. We attribute this primarily (a)
details of CWV: e.g., CWV's dependence on mixed layer temperature on the timescale of days, (b) to CWV above the mixed layer for aerosols, presumably responding to other $H_2O$ sources upwind, and (c) variations in composition: the relation of PM2.5 to light extinction. We believe that allowing for a full linear relationship each day for AOT/CWV to PM2.5, both slope and intercept effects, in a daily calibration allows regressioin to exploit portions of the PM2.5 $= f$(AOT/CWV) that isolate proportionality. High-spatial-resolution estimates of the 11 AM–
3:30 PM PBL heights for momentum may as helpful as CWV in some circumstances; this could be explored. Such PBL data is not available for the whole MODIS-Aqua period.(2004–present), while CWV is.

**Accompanying insights on pollution episodes.** We found that this approach allowed a broad description of the buildup of six air pollution episodes and the balance of the roles of accumulation of pollutants versus limited vertical mixing. Episodes were as in earlier descriptions (Watson and Chow, 2002). Each appears important in
different phases of reptitive PM2.5-increase cycles. PM2.5 to AOT relationships suggest a few days residence time for particles (actually prticulate extinction) in the Valley. The first 1–3 days after MODIS described full cloud cover could still show high, slowly decreasing PM2.5. Unpublished analysis (see Young et al. 2018) suggests that this high PM2.5 dropped preciptously when surface winds rose to > 4 m s$^{-1}$ from a quadrant centered on the NNW.

Best-estimate extensions to cloudy periods of the remote-sensing-based record can be made using the typical
meteorology of the San Joaquin or presumably other areas, and verified by extensive checks. Widely available data mapping surface winds and precipitation suffice, and do not require that detailed meteorological modeling be available.

**Role of "Static" models:** Our estimation approach aimed to avoid the use of modeling driven by source estimation and transport simulatoin. Principally we wished to provide dataseta that allowed independetn comparison
to such three-d atmospheric chemistry models (e.g., Friberg, et al., 2018). When we used RAP-model CWV rather than spatially averaged or calibratee (Just et al., 2019, manuscript in progress) MAIAC CWV, that goal was not fully reached, although RAP CWV is strongly constrained by surface, satellite, snd other observations. An aspirational goal is to provide an economical, accurate, and calibrated estimation of PM2.5 for the whole MODIS



Aqua period to date, and then beyond. The opportunities to use MISR,VIIRS, MAIA, and even geostationary imaging are appealing!

## 11. Acknowledgements;

We gratefully acknowledge the support from NASA's DISCOVER-AQ mission, followed by very encouraging continued interest and some partial support from the Health and Air Quality program management of NASA's Earth Science Applications Division. This allowed fulfilment of the prime
DISCOVER-AQ objective to demonstrate the relevance of remote sensing to specific air pollution problems. We appreciate advice individuals in that program's Applied Science Team and from the GEO-CAPE mission formulation effort (aerosol focus). Aid from Yujie Wang (NASA GSFC) regarding the MAIAC processing of MODIS data were helpful. Michael Shook's analysis suggested the use of CWV. Recent comments on the draft paper by Qian Tan and Frank Freedman are also appreciated.





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
