# Peer review of "Satellite Mapping of PM2.5 Episodes in the Wintertime San Joaquin Valley: A "Static" Model Using Column Water Vapor"

_Atmospheric Chemistry and Physics, 2019_

## Referee Comment (RC1) · Anonymous Referee #2 · 12 Sep 2019

**GENERAL COMMENTS**

The manuscript describes the derivation and application of a method that transform satellite observations into a map of surface PM2.5 concentrations. The satellite data consists of Aerosol Optical Thickness (AOT) and Column Water Vapor (CWV), both obtained from MODIS. Using a limited number of PM2.5 surface observations a statistical relation between the satellite AOT and CWV and the PM2.5 is derived, which could then be applied to an entire scene. The application is bounded to a winter season in the San Joaquin Valley, but is potentially of interest to other regions in the world too.

A novel aspect of the method is that the set of coefficients in the statistical relation is

a combination of values that are constant over the entire season (representing general valid relations), and values that are valid for a single day (to account for day-to-day varying weather conditions). The reasoning behind this formulation is explained well. Especially the step-by-step extension of the formulation illustrated by Table 1 and Figure 5 is of great help to understand the method, and provides high trust in the quality and robustness of the result. The information provided in the figures is extensive but informative, and the text provides useful description of the relevant features. The manuscript therefore deserves to be published in ACP after some minor clarifications and technical corrections.

SPECIFIC COMMENTS

The description of the mapping method mentions a number of data sources: MODIS (or specific the MAIAC retrievals of AOT and CWV), AERONET (same), the DISCOVER-AQ campaign with surface and air craft observations, EPA PM2.5 observations, and meteorological data from the RAP model. At first glance, and also at second, it is not completely clear which data is in the end used for the mapping procedure. Is that indeed just the MODIS and EPA (and campaign) surface data? To what extend does the application depend on the other data, is that just for validation, or is there also some 'hidden' usage? This could be better clarified. For example, the campaign aircraft data seems not used at all, but is mentioned already in the abstract.

Further, could some kind of physical meaning be assigned to the mapping parameters? For example, the paper clearly describes the development and presence of elevated layers of aerosol and water vapor, and how these impact the PM2.5/AOT relation. Does one of the parameters actually account for the AOTe/AOT ratio on a particular day? Not necessary for this study, but could such be validated using the aircraft observations for example?

In addition, the daily variation in aerosol concentrations and other variables is almost entirely attributed to the meteorological conditions. But what about variations in emissions? Domestic wood burning and agriculture are mentioned as important emissions sources, but these might also strongly depend on the meteorological conditions. Would strongly varying emissions have an impact on the mapping procedure, or would this simply end up in a higher RMSE? Some discussion on this would be useful.

TECHNICAL CORRECTIONS

Line 16: "Dec 11" should be "Feb 11" ?

Figure 1: For non-US citizen, could the SJV region be marked on the map? Maybe refer to Figure 6 for detailed view.

Line 137: Wrong parenthese in "(Choice" ?

Line 140: Instead of "temporal resolution", is "temporal variation" ment?

Line 142: What is "P3-B", a weather fenomena?

Line 143: Mispositioned dot in ") .AOT"

Line 234: This seems a mixture of two sentences, what is ment?

Figure 3: Time axis is unreadable.

Line 264: Which figure is ment here?

Table 1: A column with references to panels a-f in Figure 5 would be useful. Also a column with index numbers would be useful for reference. For example, line 316 could refer to the table and Figure 5 instead of "third regression".

Line 326: "The fifth estimate ...": isn't the "fourth (5d)" ment?

Line 326: "below" depends on formatting, better refer to table or figure.

Line 326: "8.03" is a different number than shown in the figure.

Line 333: "Figure 4e" should be "Figure 5e" ?

Figure 6: caption mentions "360 m agl", but figures "100 m"

Figure 6: What doe the time stamps indicate, UTC ?

Line 361: Strange sentence, should it be "Here we describe just one episode" ?

Figure 7: Caption does not correspond to figures.

Line 377: "Section 7" should be "Figure 7" ?

Line 486: "focusrd" should be "focused"

Line 514: "We attribute this primarily *to* (a) ..."

---

## Referee Comment (RC2) · Anonymous Referee #1 · 15 Oct 2019

General comments:

The authors investigated PM2.5 and AOT relationships that are affected by column water vapor (CWV) using multiple statistical model structures (including daily calibration from mixed effects model) in the San Joaquin Valley of California. As indicated by the authors, water vapor can be an important parameter to better estimate PM2.5 from AOT data because of the dry mass of PM2.5 vs. the hygroscopic property of AOT among others. It is interesting to see the authors use water vapor data retrieved along with AOT data. The authors tested multiple model structures to show the improvement of PM2.5 estimation by each model component, which is useful for readers. The San

Joaquin Valley shows high PM2.5 concentrations particularly during the winter, and therefore better understanding of PM2.5 distribution, which cannot be fully revealed by ground monitors, is important for health effect studies and air quality management. Complex terrains and meteorology (especially in winter) in the region have caused the AOT-derived estimation of PM2.5 to be a challenge, and this study investigates this critical air quality issue using a novel approach. I recommend this manuscript for publication in ACP after addressing my comments below.

Specific comments:

I suggest the authors clearly connect texts to figures and tables in the entire manuscript. I sometimes lost track of what tables or figures the texts are referring to.

Line 20: Please add the full name of rms because this is the first mention of the term.

Figure 1: This figure is based on ground observations. In the figure caption, it is better to say 'observed' or 'calculated' rather than 'estimated.'

Line 102: Please specify references.

Line 121: Satellite overpass times, 10:30 am (Terra) and 1:30 pm (Aqua), are local time not UTC.

Line 122: The period, November 2012-April 2013, is not consistent with the period mentioned in the abstract. Please also check all the periods indicated throughout the manuscript. There are lots of texts indicating study periods, but they are not always the same.

Line 143: Please clarify, RUC or Rapid Refresh (RAP)?

Figure 2: What are those three SJV PM2.5 stations? Please specify.

Line 214: This is the first mention of mixed effects model. It will be better to explain what this model is and what success the authors are referring to. Alternatively, this part

can be moved to the section after the general introduction of mixed effects model.

Line 237 (equation 5): Please use the terms, fixed slopes and intercepts and random slopes and intercepts, which are widely used. No fixed intercept included in this equation?

Figure 3(a): A boundary map of California including San Joaquin Valley is needed.

How do the colors indicate direction and distance from centroid? Also, what is the centroid here? It may be useful to include figure legends.

Line 273: Please specify what figure the authors are referring to.

Line 275: It is more useful to add references that used the aircraft data.

Figure 4: How many ground monitors are used to create this figure? There are two colored bars (dark red and light red). Does it mean there are 2 monitors for this analysis?

Table 1: The first three models do not show the full RMS error values (i.e., decimal points).

Please add a note explaining all components of the equations (e.g., i, s, a, c, alpha, and so on). The first two models need an error term and a fixed intercept.

Figure 5: For (a) and (b), values on x-axis should be indicated.

For (d), based on Table 1, R value is 0.85 and RMS error is 8.03 for this model. The authors may be confused with the fourth model in Table 1.

For (f), is this the same model as the last one in Table 1? If so, please correct the RMS error (6.48) which is indicated as 6.44 in Table 1.

Line 336: What estimate are you referring to?

Line 358 (equation 8): According to the texts, (AOT/CWV/PBL) needs to be changed to (AOT/PBL)?

Line 399: Cloud water vapor → column water vapor

Lines 431, 435, and 441: Readers do not know about specific figures included in previous research. It will be sufficient to summarize the previous research without mentioning the figure/panel numbers.

Line 446: I would remove the sentence "The whole Aqua . . . . . ." because all the MAIAC data (combined Aqua and Terra) are now available from NASA.

Figure 8 caption (lines 461 and 462): The following is true only on a given day: "There are the same for all geographical. . . The slope parameter. . . for all geographical locations." Please clearly mention it.

Line 491: Please add the full name of GOES.

---

## Author Comment (AC1) · 27 Nov 2019

We are extremely grateful for the helpful comments of both reviewers. We expect that they will find that we have addressed all comments, both general and detailed. In fact, the comments alerted us to examine and clarify several other details, for example the meaning of the intercepts and slopes as graphically analyzed, the alphas and the betas (Table 1 and Figure 8). Discussion of the comparison of the intensive Fresno measurements of Young et al. and our mapped valley-wide estimates now describe their results in some more detail. We expect that the inclusion of the Supplementary Material will clarify how the P-3B NASA aircraft studies motivated our approach. Regretfully, the

task of comparing our estimates to more detailed aircraft measurements would involve considerable additional explanation beyond the limits of a single paper. We have attempted to improve all figures up to the standards requested. Figure 1 has a more recent and descriptive source; it now indicates the position of the San Joaquin Valley.

---

## Author Comment (AC2) · 27 Nov 2019

There appeared no form to allow uploads of a revised PDF of the publication including many revisions kindly suggested but the reviewers. It is included below as a "supplement." The original supplementary material should be included without change.

Please also note the supplement to this comment:
https://www.atmos-chem-phys-discuss.net/acp-2019-262/acp-2019-262-AC2-supplement.pdf

2019.